# Chemical and Biological Studies of Endophytes Isolated from *Marchantia polymorpha*

**DOI:** 10.3390/molecules28052202

**Published:** 2023-02-27

**Authors:** Mateusz Stelmasiewicz, Łukasz Świątek, Agnieszka Ludwiczuk

**Affiliations:** 1Department of Pharmacognosy with the Medicinal Plant Garden, Medical University of Lublin, 20-093 Lublin, Poland; 2Department of Virology with SARS Laboratory, Medical University of Lublin, 20-093 Lublin, Poland

**Keywords:** liverwort endophytes, diketopiperazines, cytotoxic and anticancer activity, antiviral activity

## Abstract

Natural bioresources, predominantly plants, have always been regarded as the richest source of drugs for diseases threatening humanity. Additionally, microorganism-originating metabolites have been extensively explored as weapons against bacterial, fungal, and viral infections. However, the biological potential of metabolites produced by plant endophytes still remains understudied, despite significant efforts reflected in recently published papers. Thus, our goal was to evaluate the metabolites produced by endophytes isolated from *Marchantia polymorpha* and to study their biological properties, namely anticancer and antiviral potential. The cytotoxicity and anticancer potential were assessed using the microculture tetrazolium technique (MTT) against non-cancerous VERO cells and cancer cells—namely the HeLa, RKO, and FaDu cell lines. The antiviral potential was tested against the human herpesvirus type-1 replicating in VERO cells by observing the influence of the extract on the virus-infected cells and measuring the viral infectious titer and viral load. The most characteristic metabolites identified in the ethyl acetate extract and fractions obtained by use of centrifugal partition chromatography (CPC) were volatile cyclic dipeptides, cyclo(l-phenylalanyl-l-prolyl), cyclo(l-leucyl-l-prolyl), and their stereoisomers. In addition to the diketopiperazine derivatives, this liverwort endophyte also produced arylethylamides and fatty acids amides. The presence of *N*-phenethylacetamide and oleic acid amide was confirmed. The endophyte extract and isolated fractions showed a potential selective anticancer influence on all tested cancer cell lines. Moreover, the extract and the first separated fraction noticeably diminished the formation of the HHV-1-induced cytopathic effect and reduced the virus infectious titer by 0.61–1.16 log and the viral load by 0.93–1.03 log. Endophytic organisms produced metabolites with potential anticancer and antiviral activity; thus, future studies should aim to isolate pure compounds and evaluate their biological activities.

## 1. Introduction

*Marchantia polymorpha* L. (common liverwort, umbrella liverwort) is a common and easily cultivated liverwort species, and because of this, it is emerging as an experimental model organism for physiological, metabolic and genetic studies, as well as for evolutionary research [1,2]. Even though it is a model plant, only a few studies related to *Marchantia*-microorganism interactions have been performed [3,4,5]. 

The evolution of adaptive interactions with beneficial, neutral, and detrimental microbes was one of the key features enabling plant terrestrialization. *M. polymorpha*, a descendant of the most basal lineage of extant land plants, is gaining popularity as an advantageous model system to understand land plant evolution. However, studying evolutionary molecular plant–microbe interactions in this model is hampered by the small number of known microorganisms interacting with *M. polymorpha* [4]. In general, liverworts are resistant to the majority of potentially pathogenic microbes as they employ sophisticated mechanisms to ward off intruders [6,7]. Specialized metabolite production may be a key mechanism for protecting early land plants from environmental stresses [8]. 

Phytochemical studies on *M. polymorpha* show this liverwort produces a wide range of bioactive compounds [9,10]. The most characteristic of this liverwort species are cuparane, thujopsane, and chamigrane sesquiterpenoids, as well as marchantin-type macrocyclic bisbibenzyls. Biological activity investigations show the potential of some terpenoids and bisbibenzyls to exhibit many activities, including antibacterial, antifungal, antiviral, anti-inflammatory, and anti-proliferation of cancer cell lines. [9,11].

It should also be mentioned that endophytes associated with liverwort species can synthesize bioactive compounds, and they contribute, in part, to the control of microbial and herbivore attack or can be important for human health [12,13]. Our previous phytochemical studies on *M. polymorpha* endophytes showed diketopiperazines are the most characteristic volatile components found in the ethyl acetate extracts from microorganisms cultivated in a solid nutrient medium [14]. Diketopiperazines are the smallest cyclic peptides known and are widespread among microorganisms, especially Gram-negative bacteria [15]. These nitrogen-containing compounds have received considerable attention due to their significant pharmacological potential [16]. 

The purpose of the present studies was the phytochemical analysis of the volatile metabolites present in the endophytes of *M. polymorpha* cultivated in liquid media, isolation of the major components, and evaluation of their anticancer and antiviral activities.

## 2. Results

### 2.1. Phytochemical Studies

The ethyl acetate extract (END1) of *M. polymorpha* endophytes cultivated in BHI (brain heart infusion) liquid medium was analyzed using GC/MS. As shown in Figure 1 and Table 1, the major components detected in this extract were volatile cyclic dipeptides, cyclo(leucylprolyl) = cyclo(Leu-Pro) (**7**), and cyclo(phenylalanylprolyl) = cyclo(Phe-Pro) (**10**). Analysis of EI mass spectra showed peaks corresponding to compounds **6** and **8** have the same mass spectra as compounds **7** and **10**, respectively, but different retention indices. That means compounds **6** and **7,** as well as **8** and **10,** are pairs of stereoisomers. Based on the data published by Wang and co-workers [17], compounds **6** and **7** were identified as cyclo(d-Leu-d-Pro) and cyclo(l-Leu-l-Pro), while compounds **8** and **10** as cyclo(d-Phe-d-Pro) and cyclo(l-Phe-l-Pro), respectively. The structures of the identified compounds are presented in Figure 2.

The next step of our phytochemical studies was the isolation of proline-based diketopiperazines present in the ethyl acetate extract. To achieve this, centrifugal partition chromatography (CPC) was employed. To the best of our knowledge, there are no reports on the use of this technique to isolate compounds similar to those identified in an extract from the endophytic microorganism of *M. polymorpha*. Several two-phase solvent systems have previously been used by other authors to separate diketopiperazine compounds from marine fungi using high-speed counter-current chromatography (HSCCC) [18,19]. Both CPC and HSCCC share the same basic principles of chromatographic theory; however, the hydrodynamic retention mechanism in HSCCC caused by the varying gravitational field created by the planetary motion of the rotors differs significantly from the hydrostatic CPC, which relies on the constant gravitational interaction observed during the motion of a uniaxial rotor [20]. These differences mean there is a need to optimize chromatographic conditions while changing the technique. 

Selecting an appropriate solvent system is the most critical step for efficiently separating and purifying target compounds. The partition coefficient (*K_D_*) values obtained from the experiments described in the Materials and Methods section were used as screening parameters for selecting an optimal biphasic solvent system. Various n-hexane/ethyl acetate/methanol/water (HEMWat) mixtures, as well as the solvent system described by He et al. [19], composed of ether/ethyl acetate/methanol/water, 5.5:11:5:7 (*v*/*v*/*v*/*v* ) were tested, as shown in Table 2. 

Partition coefficient values increase as the water content of the two-phase system increases. The suitable *K_D_* values for counter-current chromatography proposed by Ito [22] should be in the range of 0.5 to 2. Based on the *K_D_* values, the solvent system composed of HEMWat, 1:19:1:19 (*v*/*v*/*v*/*v*) was selected for the isolation of the volatile cyclic peptides. 

Centrifugal partition chromatography followed by semi-preparative HPLC gave three fractions denoted as END2-END4. GC/MS analysis showed 78% of fraction END 2 was compound **10,** identified as cyclo(l-Phe-l-Pro), while compound **7,** identified as cyclo(l-Leu-l-Pro) = gancidin W, represented 83% of fraction END3. In the fraction denoted as END4, two major compounds were identified, namely *N*-phenethylacetamide and oleic acid amide, which made up 77% of this fraction (Table 1, Appendix A).

### 2.2. Cytotoxicity and Anticancer Selectivity

The cytotoxicity evaluation of the tested samples is presented in Table 3. In the case of non-cancerous VERO cells, the CC_50_ values were extrapolated from dose-response curves (inhibitor vs. normalized response) using GraphPad software (Figure 3). The ethyl acetate extract (END1) showed the lowest cytotoxicity towards VERO with a CC_50_ of 792.1 µg/mL. Subsequent fractionation (END3-END4) resulted in an increase of toxicity towards this cell line. END1 showed potentially high anticancer selectivity with an SI between 5 and 11.1. Subsequently obtained fractions END2 and END3 showed higher CC_50_ values on cancer cell lines, and a decrease in anticancer selectivity was observed. Fraction END4 showed a significantly (*p* < 0.05) higher toxicity than the initial extract and the other fractions but a lower selectivity. 

### 2.3. Antiviral Activity

The antiviral activity was evaluated against the human herpesvirus type-1 (HHV-1) replicating in VERO cells. The incubation of HHV-1-infected VERO cells with END1 200 µg/mL (Figure 4C) and END2 200 µg/mL (Figure 4E) resulted in a profound reduction of the virus-induced cytopathic effect (CPE) in comparison to the virus control (VC, Figure 4B). The effect was dose-dependent, and at 100 µg/mL of both fractions, the CPE inhibition was less noticeable (Figure 4D,F), with more cell rounding, swelling, and vacuolization and polykaryon formation. In contrast, END3 and END4 did not show any effect on the CPE (Figure 4G,H). 

Samples collected from CPE assays were further studied to assess the reduction of HHV-1 infectious titer and viral load. An endpoint virus titration assay (Figure 5) revealed the viral load was reduced by END1 (200 µg/mL) and END2 (200 µg/mL) by 0.61 ± 0.13 and 1.16 ± 0.17 logCCID_50_/mL, respectively. It could therefore be concluded that both fractions do not show significant antiviral potential since a reduction of infectious titer by at least 3 log is required. The HHV-1 infectious titers in samples treated with END3 and END4 were comparable to the VC. 

The amplification curve from real-time PCR analysis of DNA isolates from samples treated with endophyte fractions compared to the VC is presented in Figure 6A. The subsequent relative quantitation (ΔCq) method using CFX Manager Dx Software (Figure 6B) revealed END1 at 200 and 100 µg/mL decreased the viral load by 0.93 and 0.48 log, respectively. Fraction END2 at the same concentrations exerted noticeably higher inhibition, reducing the viral load by 1.03 and 0.7 log, respectively, compared with the VC. Further fractionation (END3 and END4) decreased the inhibitory effect on the HHV-1 replication. The melting curve analysis (Figure 6C) carried out after the real-time PCR amplification showed a single peak at 89.5°C for all tested samples; thus, the same amplicon was found in all samples, and no dimers of primers were observed. Acyclovir (60 µg/mL), used as a reference antiviral drug, prevented the development of CPE in the HHV-1-infected cells and the production of infectious viral progeny. Furthermore, real-time PCR amplification did not show any conclusive evidence of HHV-1 DNA in virus-infected VERO cells treated with acyclovir. 

## 3. Discussion

The evaluation of the biological activity of both bacterial and fungal endophytes isolated from various plant species provided evidence of their multifunctional activity, including antiviral and antibacterial properties [23,24,25]. Most of the published papers focus on the purification (isolation) and identification of endophyte species and then subsequently evaluate their biological activity. However, many microorganisms coexist inside plants, forming unique multidimensional interactions, including symbiosis and antibiosis not only with the host plant but also between themselves. These interactions are complex and may affect both the host and different microbes growing within the endophyte community. Obviously, when different endophyte species occupy the same environment, competition for nutrients and living space is to be expected. This may lead to the production of allelochemicals targeting other microorganisms or their toxic metabolites. However, different strategies may be encountered, including the production of secondary metabolites enhancing symbiotic relationships with other endophytes or the host they occupy [26]. Thus, we decided to not isolate particular species of endophytes from *M. polymorpha*, but to allow them to simultaneously grow in vitro, with the view that this will enable us to study the produced metabolites and assess their bioactivity. This may produce a more realistic metabolic profile observed naturally, rather than that seen for purified single endophytes. Most endophyte studies focused on their isolation from seed plants and the evaluation of species diversity, geographic location, seasonal changes, and many other parameters. However, much further effort is needed to assess the biodiversity and bioactivity of endophytes found in non-vascular bryophytes (mosses, liverworts, and hornworts) [27]. This is precisely why we have chosen to study the metabolite profile and selected biological activities of *M. polymorpha* endophytes.

Nelson and Shaw [27] reported 93 fungal isolates were obtained from *M. polymorpha* plants (subspecies *montivagans*, *polymorpha,* and *ruderalis*), representing at least 50 distinct fungal species belonging to the Ascomycota (Eurotiomycetes, Pezizomycetes, Saccharomycetes, Leotiomycetes, Dothideomycetes, and Sordariomycete) and Basidiomycota (Agaricomycetes and Tremellomycetes). Interestingly, the most abundant fungal class were Sordariomycetes, found in all *M. polymorpha* samples that yielded more than one fungal isolate [27]. Additionally, a high abundance of bacterial endophytes was observed in *Marchantia* sp., including plant growth-promoting bacteria (*Rhizobium* and *Methylobacterium*), and organic matter-decomposing bacteria (*Paenibacillus*, *Steroidobacter*, and *Lysobacter*) [28]. Our study did not include the isolation and identification of particular endophytes; however, during our initial culturing procedures using solid media [14], we observed at least several distinct types of microbial colonies growing, confirming *M. polymorpha* harbors a complex endophyte population.

Phytochemical studies on *M. polymorpha* have shown this liverwort species produce mainly sesquiterpenoids and aromatic compounds belonging to the bisbibenzyl group [9]. Extracts obtained from endophytes grown on both solid [14] and liquid (present study) media did not show the presence of plant metabolites. The most characteristic components of *M. polymorpha* endophytes are the diketopiperazines. The present studies showed the major compounds of the ethyl acetate extract are cyclo(l-phenylalanyl-l-prolyl) and cyclo(l-leucyl-l-prolyl), also known as gancidin W. In addition to these two components, their stereoisomers were also identified, cyclo(d-phenylalanyl-d-prolyl) and cyclo(d-leucyl-d-prolyl). 2,5-Diketopiperazine derivatives, also called cyclic dipeptides, are the simplest peptide derivatives in nature that are formed by the condensation of two amino acids. They are an important category of bioactive substances with various structures [29]. These compounds have recently received considerable attention due to their structural stability and significant pharmacological potential [16,30]. Cyclic peptides are widespread among endophytic microorganisms, but these are not the only metabolites found in these organisms. The current studies showed the major components of fraction END4 were *N*-phenethylacetamide and oleic acid amide. Arylethylamides and fatty acid amides are also common compounds produced by endophytic microorganisms, especially fungi. They have also been observed to be broadly bioactive against various disease agents [31,32].

The data on the biological activity of *M. polymorpha* endophyte extracts is limited. We have previously reported the cytotoxicity of extracts obtained from *M. polymorpha* endophytes cultured on Columbia agar [14]. Surprisingly, the extracts from endophytes cultured using BHI medium showed lower toxicity towards VERO, FaDu, and HeLa cells than those obtained from cultures on agar media. The ethyl acetate extract of an endophytic fungus, *C. gloeosporioides,* isolated from *Oroxylum indicum* (L.) Kurz exerted anticancer potential against HCT116 (colon cancer) and HeLa cells with CC_50_ values of 76.59 μg/mL and 176.20 μg/mL, respectively, and low activity (CC_50_—1750.70 μg/mL) against the HepG2 (hepatocellular cancer) cell line [25]. Interestingly, the *M. polymorpha* endophyte ethyl acetate extract (END1) evaluated herein showed similar activity towards colon cancer cells (71.4 μg/mL) and markedly higher activity (106.2 μg/mL) towards the HeLa cells. Our results indicate the obtained extracts potentially exert moderate anticancer potential with significant selectivity.

The observed potential anticancer activity of the endophyte extract and fractions is likely to be due to the presence of the diketopiperazines, namely l- and d-cyclo(phenylalanylprolyl) and l- (=gancidin W) and d-cyclo(leucylprolyl). This is supported by the work of Santos et al. [24] who reported that cyclo(phenylalanylprolyl) and cyclo([iso]leucylprolyl) were identified in endophytic extracts showing activity towards HepG2 and A2058 (human melanoma). Cyclo(phenylalanylprolyl) isolated from *Pheretima* sp. (earthworm) endosymbiotic bacteria was also shown to inhibit the growth of *Salmonella* Typhi and *Staphylococcus aureus*, and docking analysis revealed a potent affinity to molecular targets necessary for bacterial growth, among them dihydropteroate synthase, topoisomerase, and gyrase [23]. Notably, the DNA topoisomerases, such as type-IIA topoisomerases, are also vital therapeutic targets of anticancer drugs [33], and this may in part explain the activity observed in this study. Gancidin W was isolated from an endophytic *Streptomyces* obtained from the bark of the tree *Shorea ovalis* and showed an in vivo antimalarial effect against *Plasmodium berghei* with relatively low toxicity towards mice [34]. This compound was also the primary constituent of END3, which showed potential selective anticancer properties. The highest cytotoxicity but simultaneously the lowest selectivity to cancer cells was observed for EDN4. Interestingly, END4 did not contain any diketopiperazines; instead, the major compounds were oleic acid amide (oleamide) and *N*-phenethylacetamide. The endocannabinoid system (ECS) is a neuromodulatory system responsible for crucial processes within the central nervous system (CNS), including CNS development, synaptic plasticity, and the response to endogenous and environmental factors. It is a complex system built of cannabinoid receptors, natural ligands called endogenous cannabinoids (endocannabinoids), and the enzymatic mechanisms carrying out the synthesis and degradation of endocannabinoids [35]. Cannabinoids have a well-established role in palliative treatment in cancer patients. The endocannabinoid system still provides additional opportunities for systemic anticancer therapy by decreasing cancer cell proliferation and cancer progression, reducing neovascularization, invasion, and chemoresistance, induction of apoptosis and autophagy, as well as increasing tumor immune surveillance. Recently, several endocannabinoid-related fatty acids, such as 2-arachidonoyl glyceryl ether, *O*-arachidonoylethanolamine, *N*-arachidonoyldopamine, and oleic acid amide, were identified as endogenous cannabinoid receptor ligands [36]. Fatty acids and their amide derivatives have also been shown to target protein synthesis and cause leakage of intracellular components due to damage to the cell membranes [31]. Oleamide was shown to induce toxic effects in VERO cells, with a CC_50_ value of 33.3 μg/mL [37]. 

Endophytic organisms, mainly fungi, were also found to produce antiviral compounds [38,39,40,41,42,43]. Oryzaeins A and B, isocoumarins possessing an unusual 2-oxopropyl group and a rare 3-hydroxypropyl group, isolated from endophytic fungus *Aspergillus oryzae*, displayed moderate activity at 20 µM against the tobacco mosaic virus (TMV) with inhibition rates of 28.4% and 30.6%, respectively [42]. Versicolactone A, butyrolactone also possessing a unique 2-oxopropyl group, isolated from *Aspergillus versicolor*, showed higher inhibition of TMV—46.4% [38]. Hydroanthraquinone derivatives isolated from a culture of *Nigrospora* sp., an endophytic fungus derived from *Aconitum carmichaeli*, inhibited the replication of influenza virus (A/Puerto Rico/8/34, H1N1) with IC_50_ values ranging from 0.80 to 7.82 μg/mL [39]. Anti-influenza activity was also observed for isoindolone derivatives isolated from an endophytic *Emericella* sp. Fungus (source plant: mangrove *Aegiceras corniculatum*) [40]. Recently, aspulvinones D, M, and R, derived from a fungal endophytic *Cladosporium* sp., were found to inhibit the main protease (M^pro^) of SARS-CoV-2 with IC_50_ values between 7.7 and 10.3 μM [41]. Additionally, an ethyl acetate extract of *Curvularia papendorfii* (endophytic fungus isolated from *Vernonia amygdalina*) exerted an antiviral effect against the human coronavirus HcoV 229E and the feline coronavirus FCV F9 [44]. The antiviral potential of bacterial endophytes has however been studied to a much lesser extent, but they are also a potential source of promising antiviral compounds. For example, marine endophytic *Streptomyces* extracts showed significant inhibitory activity against the hepatitis C virus (HCV)[45]. Endophytic microorganisms are also a possible source of novel anti-HIV drugs [46]. Herein, we report the ethyl acetate extract from *M. polymorpha* endophytes (END1), as well as fraction END2, exerted low antiviral potential against the human herpesvirus type-1 replicating in VERO cells but decreased the HHV-1 infectious titer and viral load. 

## 4. Materials and Methods

### 4.1. General Experimental Procedure

The phytochemical analysis was conducted by use of the following chromatographic techniques: gas chromatography coupled to mass spectrometry (GC/MS)—Shimadzu GC-2010 Plus gas chromatograph, coupled with a QP 2010 Ultra mass detector (Shim-pol, Poland), centrifugal partition chromatography (CPC)—Gilson CPC-250 system (Middleton, WI, USA), and semipreparative high-performance liquid chromatography (semiprep-HPLC)—LaChrom 7000 HPLC system (Merck, Poland).

The evaluation of cytotoxicity and antiviral activity was conducted in a BSL-2 laboratory using a previously described methodology [47], and details can be found in the Appendix A. Cell lines included the non-cancerous VERO (ATCC, CCL-81) cells and cancer-derived cell lines: FaDu (ATCC, HTB-43, human hypopharyngeal squamous cell carcinoma), HeLa (human cervical adenocarcinoma, ATCC, CCL-2), and RKO (human colon cancer, ATCC, CRL-2577). The antiviral activity was assessed against the human herpesvirus type-1 (HHV-1, ATCC, VR-260) propagated in VERO cells. The infectious titer of HHV-1 used in the experiments was 5.5 ± 0.25 log_10_CCID_50_/mL (CCID_50_—50% cell culture infectious dose). Incubation was carried out in a 5% CO_2_ atmosphere at 37°C (CO_2_ incubator, Panasonic Healthcare Co., Tokyo, Japan). Observation of *in* vitro experiments was conducted using an inverted microscope (CKX41, Olympus Corporation, Tokyo, Japan) equipped with a camera (Moticam 3+, Motic, Hong Kong) and software for image documentation (Motic Images Plus 2.0, Motic, Hong Kong).

The evaluation of cytotoxicity and antiviral activity was conducted in a BSL-2 laboratory using a previously described methodology [47], and details can be found in the Appendix A. Cell lines included the non-cancerous VERO (ATCC, CCL-81) cells and cancer-derived cell lines: FaDu (ATCC, HTB-43, human hypopharyngeal squamous cell carcinoma), HeLa (human cervical adenocarcinoma, ATCC, CCL-2), and RKO (human colon cancer, ATCC, CRL-2577). The antiviral activity was assessed against the human herpesvirus type-1 (HHV-1, ATCC, VR-260) propagated in VERO cells. The infectious titer of HHV-1 used in the experiments was 5.5 ± 0.25 log_10_CCID_50_/mL (CCID_50_—50% cell culture infectious dose). Incubation was carried out in a 5% CO_2_ atmosphere at 37°C (CO_2_ incubator, Panasonic Healthcare Co., Tokyo, Japan). Observation of *in* vitro experiments was conducted using an inverted microscope (CKX41, Olympus Corporation, Tokyo, Japan) equipped with a camera (Moticam 3+, Motic, Hong Kong) and software for image documentation (Motic Images Plus 2.0, Motic, Hong Kong).

### 4.2. Plant Material

The fresh thalli of *Marchantia polymorpha* L. were collected between November 2020 and August 2021 from its natural state in the Lublin region (Cieleśnica, Biała Podlaska district, Poland and Strzyżewice, Lublin district, Poland) by A.L. and M.S. plant material was identified by A.L. and confirmed by Yoshinori Asakawa (Tokushima Bunri University). Voucher specimens (AL18112020-1 and MS08082021-1) have been deposited in the Herbarium of the Department of Pharmacognosy with the Medicinal Plant Garden, Medical University of Lublin.

### 4.3. Endophyte Cultivation

The isolation and cultivation of endophytes were carried out using sterile materials under aseptic conditions. The cleaned thalli of *M. polymorpha* were immersed in 70% ethyl alcohol for 60 s. Then, after firing in a burner’s flame, they were placed in 50 mL of water and shaken with sterile metal balls to homogenize and release endophytes. The obtained supernatant was inoculated in BHI (brain heart infusion) liquid medium (Oxoid, Thermo Fisher Scientific Inc, Basingstoke, UK). The endophyte isolation and culturing were carried out in triplicate; each replicate was inoculated in a total volume of 2L of BHI medium separated into glass bottles, each containing 200 mL. BHI medium was selected for culturing because it is a highly nutritious medium supporting the growth of many fastidious microorganisms. After incubation of seven days at room temperature using an orbital rotary shaker, the medium was harvested with the endophytes and subjected to the extraction procedure.

### 4.4. Extraction

The combined liquid media (6 L) with the culture of endophytes obtained from the fresh thalli of *M. polymorpha* was filtered under pressure and transferred in portions to a 1000 mL separating funnel. A liquid–liquid extraction was performed with ethyl acetate (POCH S.A, Gliwice, Poland) with three 300 mL portions. The ethyl acetate extracts were dried to remove residual water using sodium sulphate (POCH S.A), filtered and then concentrated under reduced pressure.

### 4.5. Measurement of Partition Coefficients (K_D_)

The distribution constants (*K_D_*) of each target compound in different two-phase solvent systems (*n*-hexane-ethyl acetate-methanol-water) were determined by GC/MS as follows: a small amount (10 µL) of crude extract was added into a test tube containing 4 mL of pre-equilibrated two-phase solvent system. After shaking vigorously, the mixture was left to stand at room temperature until the complete separation of the layers. Equal volumes (0.5 mL) of the upper and lower phases were taken, evaporated to dryness, redissolved in methanol (1 mL), and analyzed by GC/MS. The *K_D_* value was defined as the ratio of the peak area of a given compound in the upper phase divided by that in the lower phase.

### 4.6. Centrifugal Partition Chromatography (CPC)

Centrifugal partition chromatography was performed using a Gilson CPC-250 system (Middleton, WI, USA) equipped with a column volume of 250 mL, an Ecom TOY18DAD H (Prague, Czech Republic) detector, and a fraction collector. An optimized biphasic solvent system: hexane-ethyl acetate-methanol-water (1:19:1:19, *v*/*v*/*v*/*v*) was prepared in a glass separatory funnel to obtain appropriate volumes (1 L) of both phases. The CPC process was started by filling the column with the upper phase (stationary phase) at 500 rpm and a flow rate of 25 mL/min for 20 min. Then, the rotational speed was set at 1700 rpm, and the lower phase (mobile phase) was pumped through the column at 12 mL/min until hydrostatic equilibrium was reached. Subsequently, 380 mg of extract was dissolved in a 1:1 (*v/v*) mixture of lower and upper phases (10 mL) and injected into the system via a 10 mL loop. The entire chromatographic separation was performed in descending mode at 1700 rpm and a flow rate of 6 mL/min for 70 min. Eluates were monitored with a UV detector at wavelength 210 nm and collected into glass tubes placed in a fraction collector. Fractions of 4 mL were collected. Equal volumes (1 mL) of each fraction were taken, evaporated to dryness, redissolved in methanol (200 µL), and analyzed by GC/MS. Based on chromatographic data, fractions with similar chemical compositions were combined. Finally, five fractions (FR I–V) were obtained, which were further purified by the semi-preparative HPLC method.

### 4.7. Semi-Preparative HPLC Separation

Five fractions obtained after CPC separation (FR I–V) were dissolved in methanol (10 mg/mL) and further purified by semi-preparative HPLC, which was carried out on a LaChrom 7000 HPLC system (Merck, Poland) equipped with a D-7000 interface, an L-7150 pump, an L-7420 DAD detector, and an Advantec SF-3120 fraction collector. Separation was conducted on a Cosmosil C18-AR-II (250 mm × 10 mm, 5 µm) column with pre-column by injecting 40 μL of sample. The mobile phase consisted of 30% methanol and 70% water. The flow rate was 2 mL/min, and the fractions were monitored at 210 nm. 

### 4.8. Gas Chromatography-Mass Spectrometry 

The GC/MS analysis of the obtained extracts was carried out using a Shimadzu GC-2010 Plus gas chromatograph coupled with a QP 2010 Ultra mass detector. Volatile compounds were separated on a ZB-5MS capillary column (Phenomenex) with a silica film thickness of 0.25 µm, column length of 30 cm, and internal diameter of 0.25 mm. The initial oven temperature was 50 ^o^C, with a 3 min holding time, then increased to 250 ^o^C at 5 ^o^C/min, and held for 15 min at 250 ^o^C. An injection temperature of 280 ^o^C was maintained, and helium (1 mL/min) was used as a carrier gas. The QP 2010 Ultra mass spectrometer worked in the electron ionization (EI) mode. Ionization energy was 70 eV, the scan rate was 0.2 s/scan, and the scan range was 40–500 amu. The temperature of the ion trap was 220 ^o^C, while the injection and interface temperature were 250 ^o^C. The injection volume was 1 μL. The sample was dosed in split mode (1:20). The retention indices (RI) of the volatile compounds present in the chromatograms were calculated with respect to the homologous series of n-alkanes (C6–C27). The compounds were identified using a computer spectral library (MassFinder 2.1, NIST 2011), mass spectra of reference compounds, and available literature data.

### 4.9. Cytotoxicity and Anticancer Activity

The cytotoxicity was tested using an MTT-based protocol [47] Briefly, the cellular monolayers in 96-well plates were incubated with serial dilutions of tested extracts in the cell media (MEM or DMEM) for 72 h. Subsequently, the media was removed, wells were washed with a sterile saline solution (PBS), media with MTT was added, and the incubation continued for the next 4 h. Subsequently, SDS/DMF/PBS mixture was used to dissolve the formazan crystals, and the next day, the absorbance was measured (540 and 620 nm) using Synergy H1 Multi-Mode Microplate Reader (BioTek Instruments, Inc. Winooski, VT, USA) with Gen5 software (ver. 3.09.07; BioTek Instruments, Inc.). Collected data was exported to GraphPad Prism (version 7.04, GraphPad Software, Inc., La Jolla, CA, USA), and the CC_50_ (concentration decreasing the viability by 50%) values were calculated from dose-response curves (non-linear regression model).

### 4.10. Antiviral Activity 

The antiviral assays included the assessment of the influence of endophyte extract and fractions on the formation of HHV-1-induced cytopathic effect (CPE) in infected VERO cells, the reduction of infectious titer using the end-point virus titration, and the semi-quantitative analysis of the viral load with the real-time PCR [47]. Acyclovir (60 µg/mL) was used as a reference antiviral drug.

### 4.11. Influence on HHV-1-Induced Cytopathic Effect

The VERO cells seeded into 48-well plates were infected with HHV-1 in 100-fold CCID_50_/mL infectious dose for 1h. Afterwards, the cells were washed with PBS to remove unattached viral particles, tested extracts were added, and incubation continued until the cytopathic effect (CPE) was observed in the virus control (VC, infected, untreated cells). In every experiment, untreated and uninfected cells (cell control) were left, as well as uninfected treated cells, to observe the influence of tested extracts on the viability and morphology of VERO cells. The effect of tested extracts on the occurrence of CPE was observed and documented using an inverted microscope, plates were thrice frozen (−72°C) and thawed, and samples were collected for virus titration and viral DNA isolation.

### 4.12. End-Point Virus Titration Assay

The samples collected from the above-mentioned CPE assays were subjected to an end-point dilution assay to evaluate the HHV-1 infectious titers [47]. Briefly, the VERO cells seeded into 96-well plates were treated with ten-fold dilutions of samples in cell media for 72 h. After the incubation, the virus infectious titer for each sample was measured using the previously described MTT method. Afterwards, the difference (Δlog) between the endophyte-treated infected cells (END-T) and the virus control (VC) was calculated (Δlog = logCCID_50_VC − logCCID_50_END-T). A significant antiviral activity requires the infectious titer to be reduced by at least 3 log compared to virus control.

### 4.13. Real-Time PCR Analysis

The viral DNA was isolated using a commercially available kit (QIAamp DNA Mini Kit, QIAGEN GmbH, Hilden, Germany) according to the manufacturer’s instructions. The real-time PCR analysis was carried out with SybrAdvantage qPCR Premix (Takara Bio Inc., Kusatsu, Shiga Prefecture, Japan) and primers (UL54F—5’ CGCCAAGAAAATTTCATCGAG 3’, UL54R—5’ ACATCTTGCACCACGCCAG 3’) on the CFX96 thermal cycler (Bio-Rad Laboratories, Inc., Hercules, California, USA) [47] The viral load of HHV-1 in samples treated with endophyte extracts was assessed in relation to virus control based on the relative quantity (ΔCq) method using CFX Manager™ Dx Software (Bio-Rad Laboratories, Inc., Hercules, CA, USA).

## 5. Conclusions

All plants are capable of accommodating microorganisms termed as endophytes. Most of the research related to microbial community structure, function, and metabolism is focused on higher plants and their diversity in lower plants, such as liverworts, is neglected [48]. The last decades have seen greater attention being directed towards endophyte metabolites, mainly because of their biological potential [49]. The present studies concerning an ethyl acetate extract of *Marchantia polymorpha* endophytes cultivated in a liquid medium showed the presence of bioactive metabolites belonging to diketopiperazines, arylethylamides, and fatty acids amides. Our results indicate the crude extract potentially exerts moderate cytotoxic activity with significant selectivity. Fractions containing mainly l-cyclo(phenylalanylprolyl) and l-cyclo(leucylprolyl) showed higher CC_50_ values on cancer cell lines, and a decrease in anticancer selectivity was also observed. A fraction containing oleamide and *N*-phenethylacetamide instead of diketopiperazine derivatives showed the highest cytotoxicity but simultaneously the lowest selectivity towards the cancer cell lines. The ethyl acetate extract as well as fraction END2, with cyclo(l-phenylalanyl-l-prolyl) as the major component, exerted low antiviral potential against the human herpesvirus type-1 replicating in VERO cells. These results reinforce the potential of *Marchantia polymorpha* endophytes as a source of biologically active secondary metabolites. 

## Figures and Tables

**Figure 1 molecules-28-02202-f001:**
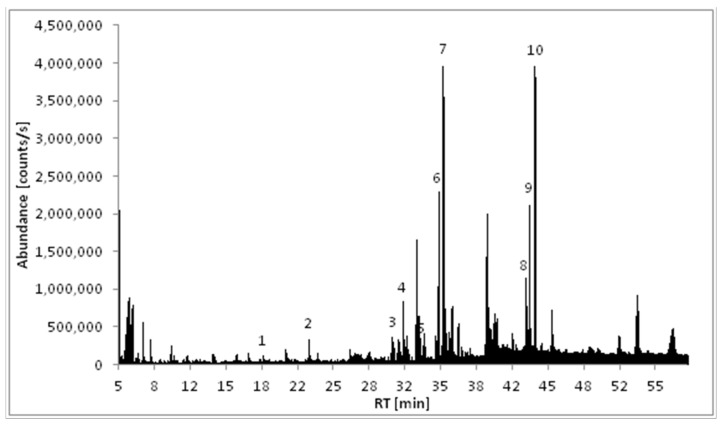
Total ion chromatogram (TIC) of the ethyl acetate extract of *Marchantia polymorpha* endophytes (END1).

**Figure 2 molecules-28-02202-f002:**
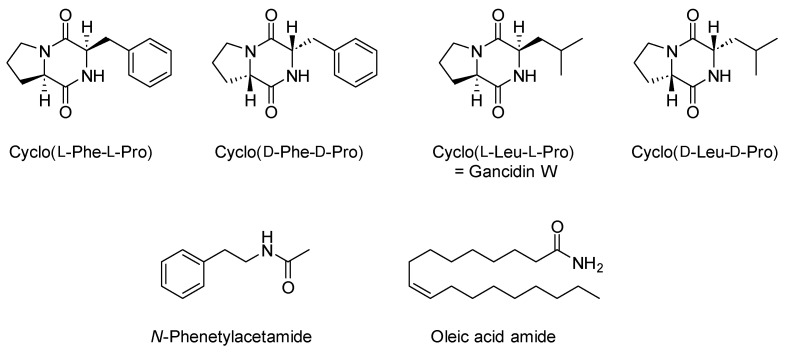
Major components identified in obtained fractions.

**Figure 3 molecules-28-02202-f003:**
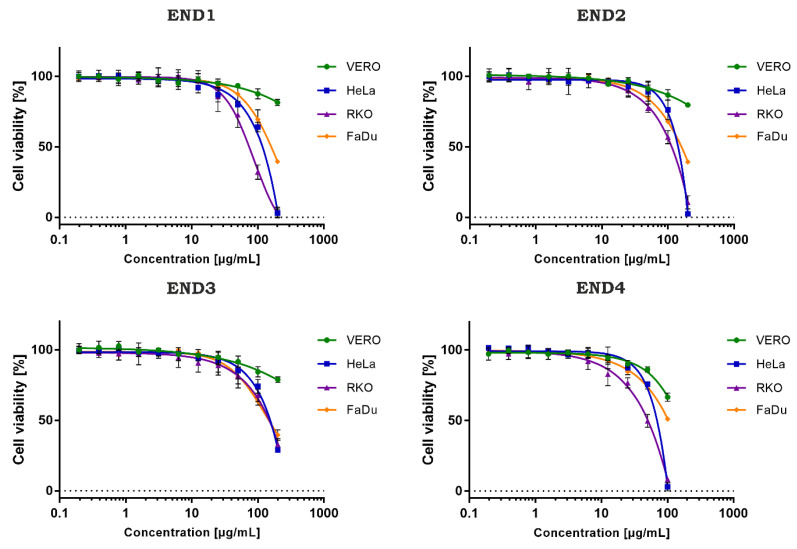
Dose-response influence of endophyte fractions (END1–4) on cell lines.

**Figure 4 molecules-28-02202-f004:**
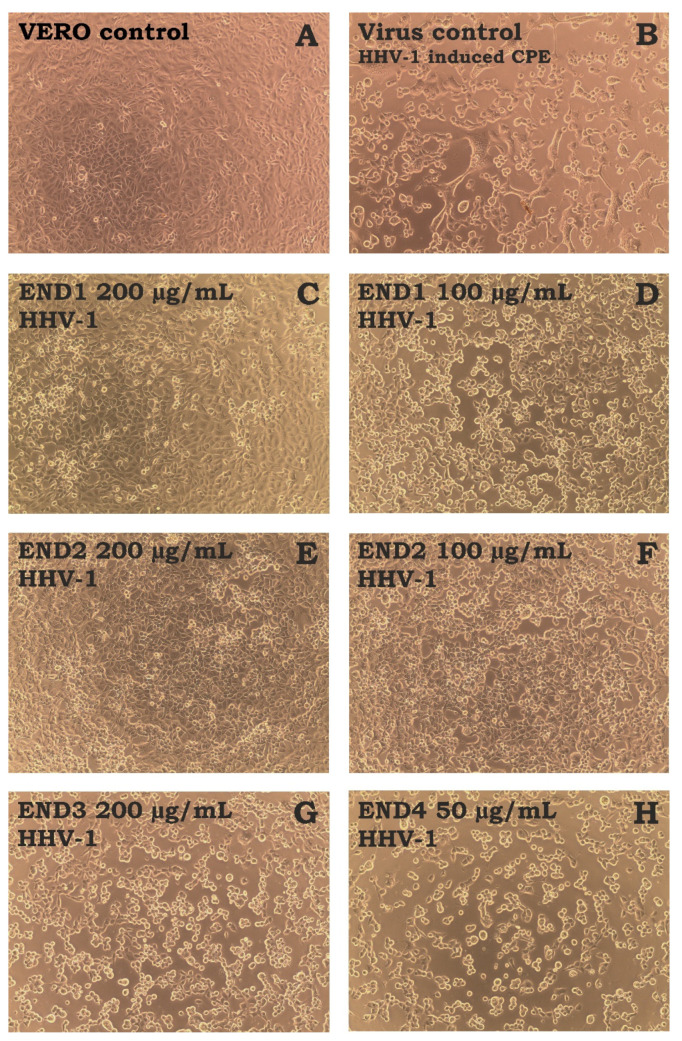
**The influence of endophyte extract and fractions (END1-4) on HHV-1-induced cytopathic effect in VERO cells** ((**A**)—cell control; (**B**)—HHV-1-induced cytopathic effect, virus control; (**C**,**D**)—infected cells treated with END1 at 200 and 100 μg/mL, respectively; (**E**,**F**)—infected cells treated with END2 at 200 and 100 μg/mL, respectively; (**G**,**H**)—infected cells treated with END3 at 200 μg/mL or END4 at 50 μg/mL, respectively).

**Figure 5 molecules-28-02202-f005:**
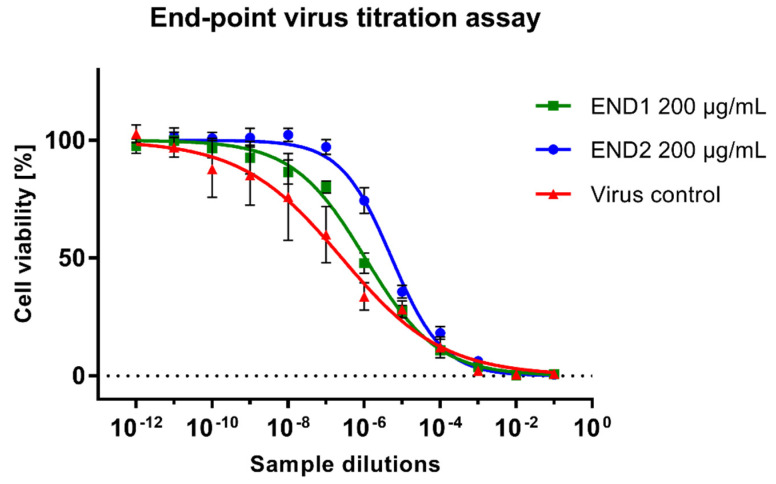
Evaluation of the HHV−1 infectious titer using the end-point titration assay.

**Figure 6 molecules-28-02202-f006:**
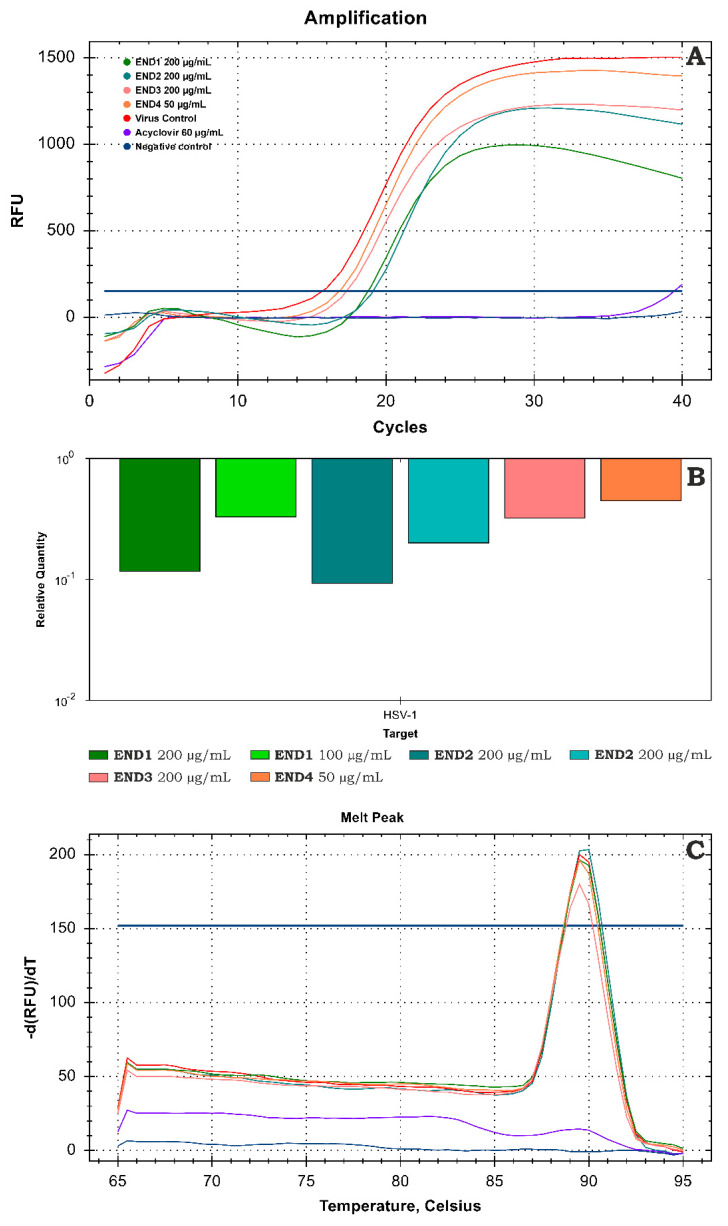
Evaluation of the HHV−1 viral load using real-time PCR (**A**)—amplification curves; (**B**)—relative quantification; (**C**)—melt curve analysis.

**Table 1 molecules-28-02202-t001:** The list of the compounds identified in the investigated samples; RI—retention index on ZB-5 column; the relative content of identified compounds was indicated as high (+++), moderate (++), low (+), or nearly (+/−) based on the peak’s surface area in corresponding chromatograms (Figures 1 and S1–S3).

No	Compound	t_R_[min]	RI	END1	END2	END3	END4
1	1,1-Dibutoxybutane	18.06	1251	+/−	+	+	+
2	Anthranilic acid	22.70	1415	+			
3	2,2-Dimethyl-*N*-phenethylpropionamide	30.40	1719	+			
4	Pyrrolidino[1,2-a]piperazine-3,6-dione	31.49	1764	+			
5	*N*-Phenethylacetamide	33.24	1848	+			+++
6	Cyclo(d-Leu-d-Pro)	34.88	1923	+		+	
7	Cyclo(l-Leu-l-Pro) = Gancidin W	35.18	1929	++		+++	
8	Cyclo(d-Phe-d-Pro)	42.95	2346	+	+		
9	Oleic acid amide	43.22	2360	+	+	+	+++
10	Cyclo(l-Phe-l-Pro)	43.81	2395	++	+++		
Percentage of major compounds in fractions					78%	83%	77%

**Table 2 molecules-28-02202-t002:** Partition coefficient (*K_D_*) values of the target diketopiperazine compounds from the endophytic organisms of *Marchantia polymorpha* L.

*n*-Hexane—Ethyl Acetate—Methanol—Water	*K_D_*				
	Compound 6	Compound 7	Compound 8	Compound 10	Reference
1:19:1:19	1.06	1.51	0.81	2.05	[21]
1:9:1:9	0.92	1.32	0.70	1.57	[21]
3:17:3:17	0.91	1.19	0.68	1.44	[21]
1:4:1:4	0.75	1.08	0.45	1.25	[21]
1:3:1:3	0.54	0.74	0.21	0.74	[21]
1:1.5:1:1.5	0.21	0.26	-	0.21	[21]
1:1:1:1	0.08	0.13	0.05	0.17	[18]
2:1:2:1	0.02	0.04	0.02	-	[18]
**Ether—Ethyl Acetate—Methanol—Water**					
5.5:11:5:7	0.31	0.42	0.15	0.37	[19]

**Table 3 molecules-28-02202-t003:** Cytotoxicity and anticancer selectivity of endophyte fractions (END1-4).

Fraction	VERO	HeLa		RKO		FaDu	
	CC_50_ *	CC_50_	SI	CC_50_	SI	CC_50_	SI
**END1**	792.1 ± 78.3	106.2 ± 8.0	7.5	71.4 ± 3.1	11.1	158.0 ± 12.0	5.0
**END2**	714.7 ± 63.3	118.4 ± 8.1	6.0	98.5 ± 4.8	7.3	155.6 ± 3.7	3.7
**END3**	646.8 ± 55.2	141.5 ± 11.5	4.6	139.7 ± 10.2	4.6	147.5 ± 15.8	4.4
**END4**	226.8 ± 14.9	54.5 ± 2.5	4.2	43.4 ± 2.4	5.2	106.1 ± 3.5	2.1

CC_50_—50% cytotoxic concentration (µg/mL (mean ± SD)); SI—selectivity index (VERO CC_50_/cancer cell line CC_50_); *—the values were extrapolated from dose-response curves (inhibitor vs. normalized response) using GraphPad software.

## Data Availability

The data presented in this study are available on request from the corresponding author.

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
