# Peer review of "Chemical and Biological Studies of Endophytes Isolated from Marchantia polymorpha"

_molecules, 2023, doi:10.3390/molecules28052202_

Round 1
Reviewer 1 Report
This manuscript investigates the chemical and biological properties of compounds derived from endophytes of the liverwort Marchantia polymorpha. It is very well written and clear.
The title starts with the word “Phytochemical” but these are generally understood as being chemical compounds produced by plants. Although not fully characterised here, the endophytes that are the source of these compounds are likely to be fungal and/or bacterial but certainly not plants so the name phytochemical should not be used here.
Throughout the manuscript it is suggested that some of the compounds from the endophytes have “anticancer selectivity”. However, this is based on the observations that the compounds are more toxic to transformed cell lines than they are to a single “normal” cell line. The promised inferred by the “anticancer selectivity” label that these compounds are anticancer agents is very premature. I recognise that many other authors use this term under similar conditions, but I believe that this is incorrect. At the very least the present authors should use the term “potential anticancer” throughout the manuscript and they should point out that only one “non-cancerous” cell line, those being Vero cells were used as a comparator. Vero cells are immortal, they are not primary cell lines and cannot really be said to be non-cancerous as being immortal is a distinct step in this direction! Generally I urge the authors to play down the inferred cancer treating promise of these compounds.
Line 215. This paragraph concentrates on fungi, but it is known that there are (naturally) a host of bacteria that exist within the thallus of Marchantia polymorpha (Alcaraz et al 2018). These include the genera Methylobacterium, Rhizobium, Paenibacillus, Lysobacter, Pirellula, Steroidobacter, and Bryobacter. As the endophytes were amplified through a seven-day culture in Brain Heart Infusion liquid medium, this will allow faster growing less fastidious organisms to outgrow other organisms present. Thus, bacteria may dominate the cultures that are the source of the compounds for these studies.
Have the authors studied these seven-day cultures to any extent? Have they looked at the cultures microscopically to see if they are predominantly fungal or bacterial?
Line 349. Can the authors provide details of the Brain Heart Infusion liquid medium? Also, what temperature were these cultures incubated and were they shaken?
In table 2. Does the lack of a given reference indicate that the source of this information is derived from the present study? If so this should be indicated.
Alcaraz, L.D.; Peimbert, M.; Barajas, H.R.; Dorantes-Acosta, A.E.; Bowman, J.L.; Arteaga-Vázquez, M.A. (2018) Marchantia liverworts as a proxy to plants’ basal microbiomes. Sci. Rep. 8, 12712.
Author Response
Response to reviewers' comments:
We would like to thank reviewer for comments on the paper, as these comments led us to an improvement of the work. Our revisions reflect all suggestions and comments. Detailed responses are given below.
Reviewer’s comment: This manuscript investigates the chemical and biological properties of compounds derived from endophytes of the liverwort Marchantia polymorpha. It is very well written and clear.
Authors’ answer: Dear Reviewer, thank you for your kind words!
Reviewer’s comment: The title starts with the word “Phytochemical” but these are generally understood as being chemical compounds produced by plants. Although not fully characterised here, the endophytes that are the source of these compounds are likely to be fungal and/or bacterial but certainly not plants so the name phytochemical should not be used here.
Authors’ answer: Dear Reviewer, thank you for pointing this out. The title has been corrected.
Reviewer’s comment: Throughout the manuscript it is suggested that some of the compounds from the endophytes have “anticancer selectivity”. However, this is based on the observations that the compounds are more toxic to transformed cell lines than they are to a single “normal” cell line. The promised inferred by the “anticancer selectivity” label that these compounds are anticancer agents is very premature. I recognise that many other authors use this term under similar conditions, but I believe that this is incorrect. At the very least the present authors should use the term “potential anticancer” throughout the manuscript and they should point out that only one “non-cancerous” cell line, those being Vero cells were used as a comparator. Vero cells are immortal, they are not primary cell lines and cannot really be said to be non-cancerous as being immortal is a distinct step in this direction! Generally I urge the authors to play down the inferred cancer treating promise of these compounds.
Authors’ answer: Dear Reviewer, thank you for your critical approach to the description of our results. We wholeheartedly agree that in vitro studies allow only for an estimation of potential anticancer activity. We also acknowledge that only one non-cancerous cell line (VERO) was used, and being a continuous cell line is not a perfect model for normal cells. Hence, wherever the results of our extracts are described, appropriate comments following your directions were included throughout the text. We have substituted cytotoxic where anticancer is mentioned in some cases to “tone down” this claim.
Reviewer’s comment: Line 215. This paragraph concentrates on fungi, but it is known that there are (naturally) a host of bacteria that exist within the thallus of Marchantia polymorpha (Alcaraz et al 2018). These include the genera Methylobacterium, Rhizobium, Paenibacillus, Lysobacter, Pirellula, Steroidobacter, and Bryobacter. As the endophytes were amplified through a seven-day culture in Brain Heart Infusion liquid medium, this will allow faster growing less fastidious organisms to outgrow other organisms present. Thus, bacteria may dominate the cultures that are the source of the compounds for these studies. Have the authors studied these seven-day cultures to any extent? Have they looked at the cultures microscopically to see if they are predominantly fungal or bacterial?
Authors’ answer: Dear Reviewer, thank you for your watchfulness. Indeed, bacterial endophytes were identified in Marchantia sp, including Marchantia polymorpha, and appropriate information was included in the modified version of the manuscript. During preliminary culturing, we performed microscopic observation of stained (Gram) preparations and observed mostly gram-positive bacteria, in shape resembling bacilli or branching rods (actinomyces?), sometimes endospore producing. However, in the presented research, our goal was to study metabolites produced by endophytes grown in co-culture, thus we have not performed isolation and identification of particular species.
Reviewer’s comment: Line 349. Can the authors provide details of the Brain Heart Infusion liquid medium? Also, what temperature were these cultures incubated and were they shaken?
Authors’ answer: During preliminary experiments, we used different media for culturing, like BHI or TSB (Tryptic Soy Broth), and we observed that cultures using BHI showed not only visually better microbial growth (turbidity and visible agglomerates) but also a higher yield of obtained extract. This may be due to the fact that the BHI medium is a highly nutritious medium supporting the growth of many fastidious microorganisms. Culturing was performed at room temperature using an orbital rotary shaker. This information was added to the revised manuscript.
Reviewer’s comment: In table 2. Does the lack of a given reference indicate that the source of this information is derived from the present study? If so this should be indicated.
Authors’ answer: Dear Reviewer, thank you for this suggestion. We have added the missing references.
Garrard, I.J.; Janaway, L.; Fisher, D. Minimising solvent usage in high speed, high loading, and high resolution isocratic dynamic extraction. J. Liq. Chromatogr. Relat. Technol. 2007, 30, 151–163
Gu, B.; Zhang, Y.; Ding, L.; He, S.; Wu, B.; Dong, J.; Zhu, P.; Chen, J.; Zhang, J.; Yan, X. Preparative separation of sulfur-containing diketopiperazines from marine fungus Cladosporium sp. using high-speed counter-current chromatography in stepwise elution mode. Mar. Drugs 2015, 13, 354–365.
Reviewer 2 Report
Dear Authors,
all comments and suggestions are in file.
Best Regards

Author Response
Response to reviewers' comments:
We would like to thank reviewer for comments on the paper, as these comments led us to an improvement of the work. Our revisions reflect all suggestions and comments. Detailed responses are given below.
Reviewer’s comment:
- Title: accurately reflects the content of the paper.
- Abstract and Key words: informative and adequate.
- Objectives and hypotheses: clearly presented.
- Methods: adequate to the aims of the study.
- Results: clear and easy to follow.
- Discussion: well-supported.
- Figures and tables: clear.
- Abbreviations, formulae, units: conform to acceptable standards.
- Literature cited: relevant.
- Presentation: good.
- Additional comments:
⎯ Interesting and aesthetic work.
⎯ The subject of this article is important. Described the Phytochemical and biological studies of endophytes isolated from Marchantia polymorpha.
⎯ Presentation and text are good.
⎯ The topic is important and original and brings new features to other publications.
Authors’ answer: Dear Reviewer, thank you for your time and appreciation of our work.
Reviewer’s comment: Please add numbers: 4.1. General Experimental Procedure; 4.2. Plant Material; 4.3. Endophyte Cultivation; 4.4. Extraction; 4.5.Measurement of Partition Coefficients (KD); 4.6. Centrifugal Partition Chromatography (CPC); 4.7. Semi-Preparative HPLC Separation; 4.8. Gas Chromatography-Mass Spectrometry; 4.9. Cytotoxicity and anticancer activity; 4.10. Antiviral activity; 4.11. Influence on HHV-1-induced cytopathic effect;4.12. End-point virus titration assay; 4.13. Real-Time PCR analysis.
Authors’ answer: Dear Reviewer, thank you for this suggestion. We have added the numbers to those sections.
Reviewer 3 Report
The manuscript addressed the phytochemical andanticancer and antiviral activities of endophytic microorganism isolated from Marchantia polymorpha.
1- The main drawback of this manuscript is the lack of experimental work of identification of the endophytes.
2- It seems there is a potential similarity between the current manuscript and the previously published work of the same group in the same journal.
(Stelmasiewicz, Mateusz, Łukasz Świątek, and Agnieszka Ludwiczuk. "Phytochemical profile and anticancer potential of endophytic microorganisms from liverwort species, Marchantia polymorpha L." Molecules 27.1 (2021): 153.)
I wonder why didn't the authors compile both work on one solid manuscript.
3- The other comments are mentioned in the attached file.

Author Response
Response to reviewers' comments:
We would like to thank reviewer for comments on the paper, as these comments led us to an improvement of the work. Our revisions reflect all suggestions and comments. Detailed responses are given below.
Reviewer’s comment: The manuscript addressed the phytochemical and anticancer and antiviral activities of endophytic microorganism isolated from Marchantia polymorpha.
Authors’ answer: Dear Reviewer, thank you for your time and suggestions that allowed us to improve our work. We have done our best to address any inconsistencies and questions.
Reviewer’s comment: The main drawback of this manuscript is the lack of experimental work of identification of the endophytes.
Authors’ answer: We acknowledge that identifying plant-associated endophytes is important, especially for environmental microbiology. However, in our work, we wanted to focus on the metabolites produced by these microorganisms growing in co-culture. This does in reality reflect a more natural situation as microbes do of course grow in co-culture. We have used a highly nutritious BHI medium in order to provide conditions supporting the growth of many fastidious microorganisms hoping this will allow us to obtain a versatile microbial growth and production of microbial metabolites. Moreover, data on the biodiversity of Marchantia polymorpha-associated bacterial endophytes can be found in the literature [Alcaraz et al., 2018], and we have included this reference in the revised manuscript. We hope that the Reviewer finds our explanations satisfactory.
Alcaraz, L.D.; Peimbert, M.; Barajas, H.R.; Dorantes-Acosta, A.E.; Bowman, J.L.; Arteaga-Vázquez, M.A. (2018) Marchantia liverworts as a proxy to plants’ basal microbiomes. Sci. Rep. 8, 12712.
Reviewer’s comment: It seems there is a potential similarity between the current manuscript and the previously published work of the same group in the same journal.
(Stelmasiewicz, Mateusz, Łukasz Świątek, and Agnieszka Ludwiczuk. "Phytochemical profile and anticancer potential of endophytic microorganisms from liverwort species, Marchantia polymorpha L." Molecules 27.1 (2021): 153.)
I wonder why didn't the authors compile both work on one solid manuscript.
Authors’ answer: Dear Reviewer, we understand your concerns and will do our best to explain our motives and rationale for separating those results. The previous report studied the endophytes cultured using agar plates and focused on the volatile compounds from Marchantia polymorpha L. and the isolated endophyte extract. During those studies, we obtained a relatively low yield of the endophyte extracts. The amount of extract was sufficient for chemical analysis and cytotoxicity evaluation but was not enough to perform other studies or isolate fractions containing bioactive compounds. That is why we have designed another study, this time using a larger-scale, liquid culture to obtain a higher yield of bacterial growth and, subsequently, a larger amount of extract. Not only this allowed for further fractionation (fractions END2-EDN4) but also to perform biological studies, including cytotoxicity and antiviral activity.
Reviewer’s comment: The other comments are mentioned in the attached file.
Authors’ answer:
- Abstract: methodology and conclusion were added
- Introduction: Common names were included - common liverwort, umbrella liverwort
- Discussion: We are sorry for our inadvertence. We have mentioned our previous paper and modified the discussion.
- Materials and Methods:
- Concerning the BHI and culturing conditions: During preliminary experiments, we used different media for culturing, like BHI or TSB (Tryptic Soy Broth), and we observed that cultures using BHI showed not only visually better microbial growth (turbidity and visible agglomerates) but also a higher yield of obtained extract. This may be because the BHI medium is a highly nutritious medium supporting the growth of many fastidious microorganisms. Culturing was performed at room temperature using an orbital rotary shaker. The endophyte isolation and culturing were carried out in triplicate; each replicate was inoculated in the total volume of 2L of BHI medium separated into glass bottles, each containing 200 mL. This information was added to the revised manuscript.
- Concerning the types of Microorganisms: As we have previously stated, we did not perform the isolation and identification of individual endophyte species. However, during preliminary culturing, we performed microscopic observation of Gram-stained preparations and observed mostly gram-positive bacteria, in shape resembling bacilli or branching rods (actinomyces?), sometimes endospore producing.
- Concerning harvesting procedure: Immediately after the incubation period, the medium was pressure filtered using a buchner funnel. Bacterial colonies were separated from a liquid medium containing endophyte metabolites. Our goal was to isolate secondary metabolites released by microorganisms into the medium. The clear medium was transferred to a separatory funnel and then the extraction process was carried out.
- Concerning the extraction procedure:
The total volume of endophyte culture was approx. 6L. In our preliminary research, we used various organic solvents such as: methanol, butanol, hexane and ethyl acetate for the extraction process. Based on the results of qualitative research on the obtained extracts, we selected ethyl acetate as a solvent for the extraction process. The ethyl acetate extract was characterized by a rich chemical composition compared to other organic extracts from endophytes.
- Concerning the methodology of cytotoxicity evaluation and antiviral studies: Experiments were done following the previously described methodology, and appropriate references were included in the revised manuscript.
Świątek, Ł.; Sieniawska, E.; Sinan, K.I.; Maciejewska-turska, M.; Boguszewska, A.; Polz-dacewicz, M.; Senkardes, I.; Guler, G.O.; Sadeer, N.B.; Mahomoodally, M.F.; et al. LC-ESI-QTOF-MS/MS analysis, cytotoxic, antiviral, antioxidant and enzyme inhibitory properties of four extracts of Geranium pyrenaicum burm. F.: A good gift from the natural treasure. Int. J. Mol. Sci. 2021, 22, 1–26
- References
We corrected the references according to your comment. All corrections are included in revised manuscript.
Round 2
Reviewer 3 Report
The authors have made an appropriate changes to improve their manuscript.